# The Asymmetric and Long-Run Effect of Financial Stability on Environmental Degradation in Norway

Dervis Kirikkaleli [1,*], Rui Alexandre Castanho [2,3], Sema Yilmaz Genc [4], Modupe Oluyemisi Oyebanji [5] and Gualter Couto [6]

1 Department of Banking and Finance, Faculty of Economics and Administrative Sciences, European University of Lefke, Lefke 99010, Turkey

2 Faculty of Applied Sciences, WSB University, 41-300 Dąbrowa Górnicza, Poland

3 College of Business and Economics, University of Johannesburg, P.O. Box 524, Auckland Park, Johannesburg 2006, South Africa

4 Department of Economics, Faculty of Economics and Administration Sciences, Yildiz Technical University, Istanbul 34210, Turkey

5 Department of Business Administration, Faculty of Economic and Administrative Sciences, European University of Lefke, Lefke 99010, Turkey

6 School of Business and Economics and CEEAplA, University of Azores, 9500-321 Ponta Delgada, Portugal

* Correspondence: dkirikkaleli@eul.edu.tr

**Abstract:** Risks associated with climate change can have an injurious impact on the economy as well as the financial system as a whole. There is a possibility that certain risks, such as losses to financial intermediaries and disruptions in the functioning of financial markets, can aggravate vulnerabilities in the financial system under certain conditions, including sudden increases in the prices of large asset classes. Using the dataset for Norway between 1995 and 2018, this study investigates how financial stability affects environmental degradation in Norway while controlling openness in trade, ecological clean energy, and economic growth. Findings from the results demonstrate that (i) financial stability causes a reduction in environmental degradation; (ii) growth causes carbon emissions to climb significantly; and (iii) renewable energy has been favorable for emissions in Norway. Lastly, surprisingly, trade openness causes a decline in carbon emissions. The study recommends that since financial stability in Norway reduces environmental degradation by incorporating climate-related risks into the financial stability monitoring framework, it can contribute to lowering carbon emissions to a greater extent. Norway's policymakers should conduct detailed analyses of the role of global emissions in long-term petroleum policy and the economic viability of selected climate policy scenarios before implementing such a policy. Moreover, policymakers should be updated on the financial system's vulnerabilities, considering climate-related shocks are likely to affect all financial systems. In addition, policymakers should encourage the use of sustainable energy to raise the availability of reliable, affordable, and sustainable energy to everyone.

**Keywords:** financial stability; environmental degradation; economic growth; trade; Norway

## 1. Introduction

Since the 1970s, Norway has become one of the world's largest sovereign wealth funds, with USD 1 trillion in assets. Although Norwegians consume 90% of their domestic power through hydroelectric power, the transportation and oil production sectors mostly use oil. The country suffered a major crisis in the 1990s which led to the loss of equity capital and market freeze-ups for several large banks. There were several banks that went bankrupt and lending rates were extremely high. However, Norway's economy has expanded and its economy has recovered. According to [1], the country has one of the highest GDPs per capita in the EU. Futher, Norway ranks third among global exporters of natural gas and 15th among exporters of crude oil. Norway is a modern country, rich in renewable

energy, and is among the world's wealthiest. Having a strong external financial position, Norway has a lot at stake in promoting a market-based economy [1]. A stable financial and economic environment may facilitate increased investments and production in non-renewable sources of energy, which can trigger $CO_2$ emissions ($CO_2E$) on a large scale. This might be true for a country such as Norway both in the present and in the future. To add to this argument, futuristically, for a country such as Norway, global warming and changes in the ecosystem due to the rise in greenhouse gas (GHG) emissions can reduce the quality of the environment, which might have a ripple effect on all sectors that sustain the economy. Emissions stem from several factors, such as urban growth, the rise in population growth, and the utilization of fossil fuels for domestic, high technological progress and industrial expansion, which can disrupt the economy if not cautioned [2]. What is more, Norway, being the largest producer of fossil fuels in the EU, has the ability to increase its emission level (because of the Russia/Ukraine war). Many EU countries are trying to reduce energy reliance on Russia. Considering that energy is both a contribution (input) in the process of production and an end-user product, reducing its use to promote sustainability will stifle progress in both human and economic development. Due to the preceding debates, two important questions emerge: (1) Does Norway's financial stability increase or decrease $CO_2E$?; (2) How green will Norway be in the near future? As of 2021, Norway had produced 33.4 million metric tons of carbon dioxide ($MtCO_2$) emissions. The increase has been about 1.5 percent year-over-year and is greater than 15 percent from 1990. The goal of this empirical work is to capture the effect of financial stability on environmental degradation in Norway while controlling economic growth, renewable energy, and free trade. Our research study contributes to the literature by jumping into the debates on financial stability, renewable energy consumption, and $CO_2E$. The threat of climate change poses an existential threat to humanity. Climate scientists are sharing a serious warning to societies, governments, and policymakers that ineffective current policies should be changed before extreme weather events become more destructive and more frequent, and global warming becomes more disruptive. When environmental conditions worsen, this can contribute to dramatic declines in the valuations of fossil-based energy companies, stock market panics, and an increase in the number of insurance payouts for losses and damages resulting from extreme weather events, drought, and flooding.

Norway, a country with a population of 5,497,878 million and a GDP per capita of USD 6,729,448 [3], has much of its economic activity centered around international trade and is reliant on natural resources (including hydropower, natural gas, petroleum, etc.). However, oil and natural gas are major resources. Over the years, exports of oil and gas have been rising, and industries that require energy are growing as well. Norway's economy is heavily reliant on crude oil and natural gas production; oil and natural gas exports make up two-thirds of its GDP and 22% of its exports [4,5]. In its south and central regions, fishing and related industries provide employment in coastal settlements, while forestry supports rural employment. International trade plays a significant role in Norway's economy. About three-quarters of Norway's oil supply is exported to EU nations, making up about 2% of the global oil supply. There has been awareness for a long time of domestic and international environmental issues in Norway, and the country is exposed to air pollution and coastal water pollution, as it is a very large exporter and producer of fossil fuels [5]. There are a number of environmental concerns in Norway, including climate change, biodiversity, acidification, eutrophication, toxic contamination, and hazardous waste management. Although Norway's electricity and heating needs are largely met by hydropower, emissions from transportation and oil and gas extraction are on the rise. Almost a third of Norwegian greenhouse gas emissions are attributed to the transport sector. Over one-quarter of Norway's domestic $CO_2E$ comes from the production of fossil fuels [6–8]. Yet, Norway is poised to be one of the leaders leading climate action worldwide. Given these issues, we wonder whether continued production and exportation of crude oil and natural gas will outweigh efforts to reduce domestic emissions. Norway aims to lead climate action globally. However, will continued crude oil and natural gas production outweigh Norway's efforts

to reduce domestic emissions? Will the climate policies implemented be consistent with the Paris agreement to decrease worldwide demand for oil and gas and make investments in their future production unprofitable? Global climate targets should be evaluated in light of these potential risks in order to economically stabilize the usage of fossil fuels for nonrenewable energy production.

Norwegian banks consist of commercial banks, regional banks, and small savings banks. Norwegian banks are generally well capitalized and have high leverage ratios, so they are relatively resilient to a large increase in loan losses. In addition to overseeing the entire financial system, Norges Bank is responsible for ensuring that all institutions within the financial system comply with current regulations. Furthermore, the Norwegian Financial Supervisory Authority imposes new regulations on practices within the financial system or issues recommendations. Further, the Norwegian Ministry of Finance has an important coordinating role should a financial crisis arise. By lending funds to banks against approved collateral, Norges Bank can contribute to a liquidity shortage. It is imperative for the smooth functioning of an economy that the financial sector generates funds, facilitates transparent transactions, and manages capital. It is crucial to maintain a robust financial and economic sector for the effective redistribution of resources and to heighten the productivity of business operations, which as a consequence results in higher economic expansion. Additionally, a sound financial system offers more investment opportunities by making financial goods and credit easier to obtain at low interest rates, allocating resources for developing industries, and promoting trade. Aimed at highlighting the importance of financial transactions in economic prosperity, Schumpeter was the first to assess the significance of financial transactions on economic expansion. Following this, several other research studies have reported that an advanced financial structure strengthens the economy. As the financial system plays an important role in an economy and has a significant impact on the quality of the environment, numerous studies have been conducted. However, their findings are contradictory [9,10]. The threat of climate change poses an existential threat to humanity. Climate scientists are sharing a serious caution. We must change our ineffective current policies before extreme weather events become more destructive and frequent and global warming becomes more disruptive [11–13].

The financial sector is likely to facilitate eco-sustainable initiatives with symmetrical knowledge in the event of financial difficulties [14–16]. The absence of symmetrical knowledge harms businesses that perform well and can trigger many challenges. Hence, it will be difficult for businesses to pay back their loans swiftly, especially because ecological imbalance can affect the ecosystem at large. Further, the surge in the atmosphere and other economic and financial misfortunes that have been reduced by climate change could lessen businesses' profitability, consequently weakening their financial positions. Hence, defaults on debts will increase which could result in systemic risk, similar to the Norwegian banking crisis in the 1990s. In addition, continuous firm profitability coupled with the climate crisis can trigger investors' confidence, resulting in an increase in liquidity preference. To our knowledge, no research studies on the effect of financial stability on $CO_2E$ in Norway have been conducted up to now. This study is one of the first to examine the linkage between financial stability and environmental degradation, using the NARDL, Fourier, and nonlinear cointegration approaches and other econometric techniques. This section provides a synopsis of the literature. Part 3 covers various econometric methods and data sources. The subsequent section presents empirical findings and provides an interpretation of the study under review. The last section presents the conclusion and policy implications.

Historically, economists have acknowledged that a stable financial system is crucial to sustainable output growth [17]. As financial systems develop, savings are organized better, capitalists are created more efficiently, financial risks are diversified, and credit is distributed more consistently among potential borrowers [18]. Financial sector improvements lower hedging costs, reduce loan costs, expedite FDI-related inflows of funds, and supervise the activities of firms. Further, as [19] has argued, financial industry development is fundamental for decreasing loan transaction costs, as well as reducing information asymmetry.

According to the empirical growth literature over the last few decades, there has been a persistent relationship between finance and growth [20,21]. As previously noted, a sound financial system is considered a prerequisite for better environmental quality, in addition to its importance for long-run economic progress. While the goal of economic development and environmental quality did not receive ample attention in the growth–environment research studies, recent studies have illustrated the paucity of information on this link [22,23]. Research conducted thus far is limited to the empirical testing of [24] the environmental Kuznets curve theory. According to this view, environmental damage occurs at the early stages of economic development because natural resources are overutilized, and energy is polluted. When economic development has reached a certain point, the availability of resources and increasing public pressure to keep the environment clean will force policymakers to adopt clean energy technologies. Various debates have suggested that the inconsistent outcomes of past investigations shed light on the reality that nations' emissions levels are not solely influenced by their incomes but also by other important factors and components of their socioeconomic profiles that regulate the relationship between energy and $CO_2E$ [25]. There are various macroeconomic variables that can be linked directly or indirectly to pollution emissions, as demonstrated by the recent literature. Regarding the significance of the financial sector, the studies of [26] explain that financial progress fosters strong governance among firms, which leads to superior environmental quality. In contrast, the authors of [27] opined that the expansion of the financial system advances economic boom and industrial expansion, leading to environmental degradation. However, as noted in the literature, there still remains a gap regarding the nexus between financial development and $CO_2$ emissions, in which some authors claim that the variables are negatively related [28], while others argue a positive relationship [24,29,30]. Likewise, another study finds no evidence of a finance–emissions relationship for the sample nations [29,31]. Some studies suggest that the resilience and stability of the financial system and not just the degree of financial intermediation is what affects environmental quality. To understand how financial stability impacts pollution in the environment, it is necessary to test the direct connection between these two avenues rarely explored in empirical research. It has been reported that financial instability negatively impacts environmental quality. A handful of studies have delved into financial instability and carbon emissions. However, little or no studies have sought after the nexus between financial stability and carbon emissions. For instance, the authors of [32] pointed out that financial crises did not precipitate environmental pollution. The study of [33] showed a direct link between financial instability and economic stagnation. Further, the study revealed that financial development promotes good governance among firms and therefore results in better environmental quality. The result of [34] confirmed that financial stability is a key driver for $CO_2E$ in China. According to the authors of [35], who explored the favorable and unfavorable interactions between financial instability and $CO_2$ emissions in Turkey, the results showed the long-term impact of financial stability on $CO_2$ emissions is not highly significant, using annual data from G7 advanced countries (Canada, France, Germany, Italy, Japan, the United Kingdom, and the United States). Employing the fixed effect model as well as some other econometric techniques, the findings of the study reveal that climate change (carbon emissions) increases sovereign risk significantly. Transport, electricity, and industry all contribute substantially to environmental pollution and contribute significantly to sovereign risk, as demonstrated in this study.

Further, the use of renewable energy sources is one of the alternative measures contributing significantly to improving environmental quality. As a result of renewable sources of energy, energy can be naturally produced to meet domestic energy requirements without damaging environmental quality [36]. Researchers have begun to study the relationship between alternative energy sources and the intersection of environmental quality, taking into account the value of renewable sources of energy. Using data from six Central American nations from 1974 to 2004 [37], researchers questioned the linkages between sustainable energy sources and ecological sustainability for the first time. Findings have revealed

that the usage and production of sustainable energy sources reduce $CO_2E$. A study [38] examined the effect of producing and using renewable energy sources on the economy of the Republic of China and poor air quality from 2000 to 2011. The study indicated that the production and use of renewable energy sources had a positive influence on the economy and air quality. Similarly, the authors of [39] applied advanced quantile modeling to analyze the association between the energy utilization from some renewable sources and environmental degradation from 1990 to 2017. It is clear from the study's findings that renewable energy use and environmental protection are bi-directional. Further, there have been studies on the effect of sustainable and nonrenewable energy use on emissions in various regions and countries, although the results have been mixed. For example, the authors of [40] used a variety of metrics to explore how renewable energy mitigates pollution and energy intensity, whereas nonrenewable energy exacerbates these characteristics in thirty-four (34) Sub-Saharan African countries. Both renewable energy and the use of renewable energy have mitigatory effects on emissions. Ref. [41] report that renewable energy decreases pollution while nonrenewable energy aggravates it in twenty (20) emerging countries. Renewable energy has a mitigating effect on environmental degradation, while nonrenewable energy negatively impacts ten African countries, according to [42]. Further, utilizing macroeconomic factors, such as energy consumption and economic growth, it has been argued for a long time that trade openness and carbon emissions are related, and it is an important issue in trade policy. Several studies have looked at trade openness and its impact on environmental quality, but they have found mixed results. Some have claimed that trade openness improves environmental quality, whereas many others have found that there is no connection between the two. For example, the authors of [43] studied the interrelationship between trade and environmental factors in sixty (60) advanced and emerging economies between 2002 and 2012. They found that the environmental performance index improves when trade openness is taken into account as part of a fixed effect and GMM model. The GMM model shows that political factors enhance environmental quality, whereas population and income negatively impact it. The authors of [44] have studied the impact of trade openness on the degradation of the environment in five Asian countries from 1995 to 2014. For their sample countries, they found long-term associations between the variables. The findings also confirmed the bidirectional links between carbon emissions, energy consumption, and economic growth. The EKC hypothesis was supported by [45] in their study of twenty-seven developed countries.

## 2. Methodology

Financial stability, renewable energy, and free trade are taken into account as factors that affect $CO_2E$ in Norway over the period of 1995 Q1–2018 Q4. The fact that Norway's Climate Change Act came into force in 2018 and a significant surge in $CO_2E$ in Norway led to the choice of 2018. Choosing these dates was a careful decision for this study because it seeks to present a comprehensive analysis of what significant changes have been made between the point at which emissions gained a significant increase and the date the Climate Change Act was introduced.

A nation's economic growth and prosperity depend on the stability and development of its finances. Financial development implies the growth and evolution of the financial sector, exemplified by an increased monetization of the economy, increased scale of financial institutions, and financial innovation. A nation's economic growth and prosperity are dependent on financial stability and development. In order to achieve financial development, the growth and development of the financial system are crucial for an economy's prosperity. In the real economy and society, the importance of the financial sector has been recognized for centuries [46]. In addition to its importance, the financial sector has a crucial role to play. Through its financial markets and institutions, the government redistributes resources from savers, such as households or institutional investors, to firms. Investing in the former will produce a profit, whereas investing in the latter will require financial resources to proceed [47]. The financial development of a firm influences its ability to invest. Norway's

financial degradation brought rapid credit expansion in the 1980s. As a major oil exporter, Norway enjoys substantial financial buffers. Norway's abundance of natural resources accounts for a significant part of its wealth. Through the utilization of hydropower resources and the development of large metal, chemical, and paper mills, industrialization took off in the early 20th century. Since Norway discovered large offshore deposits of petroleum in the late 1960s and onwards, the country has become one of the world's leading energy exporters. Petrochemical discoveries are still being made economically, and large parts of the continental shelf have not yet been explored. Moreover, the Norwegian financial system is fairly stable, capable of managing resource allocation efficiently, assessing and managing risks, and maintaining employment levels around the economy's natural rate. We establish that Norway's economy is financially stable; hence, we explore the effect of financial stability on environmental degradation. The predominance of continuous growth in the economy has resulted in a steady increase in pollution in many nations, prompting governments to implement changes in the energy basket of their economies. However, this change involves an increase in technology in these countries, and therefore, the ongoing reliance on the fossil fuel-based manufacturing process must be minimized while supporting the development of renewable energy. Several EU countries have replaced Norwegian oil and gas obtained from Russia with their own clean energy, which may result in an increase in fossil fuel prices, noting that Norway is one of the world's biggest oil and gas producers, contributing significantly to European energy security. Among the implications of this study is that it assists Norway and other oil-producing countries to maintain their emissions targets based on the climatic objectives of 2050 even when there are opportunities to deviate from it.

Trade openness is also a powerful tool for increasing economic growth, creating jobs, and reducing poverty. Trade benefits domestic firms by creating new sales opportunities, increasing productivity, and fostering innovation [48]. As the global economy grows faster due to free trade, both developed and developing countries see increased trade volume and income. However, the environment also suffers because of this growth trend. Since Norway is an import–export economy, the effect of trade openness on the environment will be significant; hence, adding trade openness to our paper will be useful. Further, renewable technologies improve energy security; therefore, energy policies should be promoted to reduce emissions and increase energy efficiency. Renewable energy is another driver for this reduction. Thus, we introduce renewable energy in our study. Finally, economic growth implies increased energy use, which feeds the cause of $CO_2E$, so pollution directly relates to growth and development. Every country wants to increase economic growth, but how will this affect the level of emissions in Norway? This study attempts to provide an answer for the question.

$$CO_{2Et} = \vartheta_0 + \vartheta_1 FRI + \vartheta_2 EGR_t^- + \vartheta_3 RENN + \vartheta_4 TRO + \varepsilon_t \tag{1}$$

**H1.** *Financial stability negatively influences $CO_2E$.*

We predicted that FRI would play a beneficial role in carbon emissions and we projected that the effect of FRI on $CO_2E$ is negative, i.e., $\tau_1 = \frac{\partial CO2E}{\partial FRI} < 0$. In a stable financial environment, resources are allocated efficiently. Risks are assessed and managed, employment levels are low, and less $CO_2E$ is produced. This implies that a stable financial system is necessary to invest, grow, and be involved in research and development so as to produce a sustainable environment that reduces carbon emissions. The findings of [49–51] are in line with this expectation. The following are the projected signs of the determinant of environmental degradation in Norway.

**H2.** *There is a positive relationship between EGR and $CO_2E$.*

Moreover, we expect a positive association between EGR and environmental degradation, $\tau_2 = \frac{\partial CO2E}{\partial EGR} > 0$ [52].

**H3.** *There is a significant relationship between renewable consumption and CO$_2$E.*

In addition, we expect to capture the negative effect of RENN on CO$_2$E in Norway, i.e., $\tau_3 = \frac{\partial CO2E}{\partial RENN} < 0$. This is because, unlike fossil fuels, renewable energy sources do not release greenhouse gases into the atmosphere in the process of generating electricity [53–55].

**H4.** *Trade openness increases carbon emissions.*

Finally, the present study expects trade openness to have a positive effect on the environment. This is because imports and export are vital elements of economic growth, which in turn leads to higher CO$_2$E in Norway [56,57].

## 3. Empirical Finding and Discussion

This study investigates how financial stability affects environmental degradation in Norway while controlling openness in trade, ecological clean energy, and economic growth. The present study used Fourier ADL cointegration, nonlinear ARDL, and frequency domain causality tests to capture the effect of financial stability on environmental degradation in Norway. In Table 1, we provide an overview of the study series. In Table 1, the highest mean is TR (11.26344), followed by FRI (3.817371), while the lowest is CO$_2$E (1.639571). All variables are negatively skewed; all the variables were in line with normal distribution based on the kurtosis outcomes. The study focuses on the asymmetric effects of financial risk, growth, sustainable energy use, renewable energy, and trade openness on carbon emissions between 1995 and 2019. A statistical summary and the different normality tests used are presented in Table 1. Further, the parameters' mode, maximum, mean, standard deviation, minimum, and median are depicted by the descriptive statistics. Table 1 shows that GDP has the highest mean (11.52861), thereafter followed by trade openness (TR) with a mean of 11.26344, while FRI, RE, and CO$_2$E have means of 3.81737, 11.823798 and 1.639571, respectively. According to the research, the series were tested for light-tailedness or heavy-tailedness in relation to a normal distribution by applying kurtosis.

**Table 1.** Descriptive Statistics.

|  | CO$_2$E | FRI | EGR | RENN | TRO |
|---|---|---|---|---|---|
| Mean | 1.639571 | 3.817371 | 11.52861 | 1.823798 | 11.26344 |
| Median | 1.644849 | 3.828641 | 11.54582 | 1.825861 | 11.27240 |
| Maximum | 1.661848 | 3.888413 | 11.60989 | 1.861317 | 11.34623 |
| Minimum | 1.566293 | 3.569533 | 11.39621 | 1.789666 | 11.10469 |
| Std. Dev. | 0.016813 | 0.056395 | 0.056913 | 0.014033 | 0.052989 |
| Skewness | −1.533885 | −2.180503 | −0.506929 | −0.188092 | −0.853525 |
| Kurtosis | 6.688577 | 9.357554 | 2.315771 | 3.169775 | 3.555202 |
| Jarque-Bera | 95.90339 | 247.6536 | 6.233655 | 0.709744 | 13.42612 |
| Probability | 0.000000 | 0.000000 | 0.044297 | 0.701263 | 0.001215 |

As part of this study's econometrics approach, we employed the BDS test, and the purpose of this test is to affirm whether linear or nonlinear modeling is appropriate for the study. Hence, the research employed the commonly used BDS test. Shown as Table 2.

Table 3 displays the BDS-based outcomes for the time series variables. Clearly, Table 3 demonstrates that the BDS test has very robust conclusions, and all the statistics reject the null hypothesis. If the traditional linear framework is used for analysis, significant deviations may result in biased predictions of carbon emission.

**Table 2.** BDS Test.

| | CO$_2$E | | | |
|---|---|---|---|---|
| Dimension | BDS Statistic | Std. Error | z-Statistic | Prob. |
| 2 | 0.168316 | 0.007737 | 21.75403 | 0.0000 |
| 3 | 0.283795 | 0.012368 | 22.94651 | 0.0000 |
| 4 | 0.359338 | 0.014812 | 24.26011 | 0.0000 |
| 5 | 0.406884 | 0.015526 | 26.20583 | 0.0000 |
| 6 | 0.438087 | 0.015059 | 29.09110 | 0.0000 |
| | FRI | | | |
| Dimension | BDS Statistic | Std. Error | z-Statistic | Prob. |
| 2 | 0.124591 | 0.009201 | 13.54141 | 0.0000 |
| 3 | 0.200905 | 0.014712 | 13.65612 | 0.0000 |
| 4 | 0.251135 | 0.017629 | 14.24556 | 0.0000 |
| 5 | 0.276599 | 0.018492 | 14.95787 | 0.0000 |
| 6 | 0.283626 | 0.017949 | 15.80216 | 0.0000 |
| | EGR | | | |
| Dimension | BDS Statistic | Std. Error | z-Statistic | Prob. |
| 2 | 0.206274 | 0.005371 | 38.40511 | 0.0000 |
| 3 | 0.350489 | 0.008575 | 40.87364 | 0.0000 |
| 4 | 0.451995 | 0.010253 | 44.08257 | 0.0000 |
| 5 | 0.524054 | 0.010729 | 48.84303 | 0.0000 |
| 6 | 0.575553 | 0.010387 | 55.40981 | 0.0000 |
| | RE | | | |
| Dimension | BDS Statistic | Std. Error | z-Statistic | Prob. |
| 2 | 0.143721 | 0.007774 | 18.48745 | 0.0000 |
| 3 | 0.226231 | 0.012448 | 18.17460 | 0.0000 |
| 4 | 0.269320 | 0.014933 | 18.03506 | 0.0000 |
| 5 | 0.285611 | 0.015680 | 18.21456 | 0.0000 |
| 6 | 0.285535 | 0.015234 | 18.74269 | 0.0000 |
| | TR | | | |
| Dimension | BDS Statistic | Std. Error | z-Statistic | Prob. |
| 2 | 0.201897 | 0.007630 | 26.45935 | 0.0000 |
| 3 | 0.343007 | 0.012182 | 28.15675 | 0.0000 |
| 4 | 0.442892 | 0.014572 | 30.39429 | 0.0000 |
| 5 | 0.513316 | 0.015256 | 33.64775 | 0.0000 |
| 6 | 0.563869 | 0.014778 | 38.15610 | 0.0000 |

**Table 3.** Unit Root Test.

| | | At Level | | | | |
|---|---|---|---|---|---|---|
| | | CO$_2$E | FRI | RENN | TRO | EGR |
| LS | t-Statistic (tau) | −4.1304 | −5.4503 | −4.8015 | −5.2641 | −4.071412 |
| | Break Points | 2000Q4 2008Q4 | 2008Q4 2015Q3 | 1999Q3 2004Q3 | 1998Q3 2008Q4 | 2004Q1 2009Q3 |
| | 1% level | −6.9320 | −6.932000 | −6.7500 | −6.9320 | −6.821000 |
| Test critical values | 5% level | −6.1750 | −6.175000 | −6.1080 | −6.1750 | −6.166000 |
| | 10% level | −5.8250 | −5.825000 | −5.7790 | −5.8250 | −5.832000 |
| | | At First Difference | | | | |
| | | CO$_2$E | FRI | RENN | TRO | EGR |
| LS | t-Statistic (tau) | −6.7717 *** | −10.839 *** | −6.2995 ** | −7.1696 *** | −6.5129 ** |
| | Break Points | 1997Q3 2016Q3 | 2000Q1 2015Q1 | 1999Q3 2003Q3 | 2002Q3 2007Q3 | 2003Q3 2011Q1 |
| | 1% level | −6.8210 | −6.8210 | −6.7500 | −6.9320 | −6.9780 |
| | 5% level | −5.9170 | −5.9170 | −6.1080 | −6.1750 | −6.2880 |
| | 10% level | −5.5410 | −5.5410 | −5.7790 | −5.8250 | −5.9980 |

Note: ** and *** denote statistically significant at the 5%, and 1% levels, respectively.

The current research makes a further contribution to this body of knowledge by assessing the stationarity of the variables. The LS unit root test was used in order to determine the characteristics of time series variables. Table 3 summarizes the results of the LE unit root test conducted by [58] which clearly signal that the LCO2, LFRI, LEGr,

LRENN, and LTRO are not stationary at levels. However, after taking the first difference in the time series variable, the time series variables become stationary. The breakpoints of the time series variables are: $CO_2E$ (1997Q3; 2016Q3), financial risk (2000Q1; 2015Q1), RENN (1999Q3l; 2003Q3), TRO (2003Q3; 2007Q3), and lastly EGR (2003Q3; 2011Q1).

In the next step, we employed the Fourier ADL cointegration and the nonlinear ARDL bound tests to capture whether the combination of FRI, RENN, TRO, and EGR significantly effects $CO_2E$ in the long run in Norway. Table 4 depicts the results of the Fourier ADL cointegration and the nonlinear ARDL bound tests. In addition, the outcome of the tests demonstrates that there has been a long-term association between $CO_2E$, FRI, RENN, TRO, and EGR for Norway. The outcomes allow us to capture the effect of FRI, RENN, TRO, and EGR on $CO_2E$ in Norway using the nonlinear ARDL approach.

**Table 4.** Fourier and Nonlinear Cointegration Approaches.

| Fourier ADL Cointegration Analysis | | | | |
|---|---|---|---|---|
| Model | Test Statistics | Frequency | Min AIC | |
| $CO_2E = f(FRI, EGR, RE, TRO)$ | −5.772 *** | 2 | −8.661645 | |
| Nonlinear ARDL Bounds Test | | | | |
| F-Bounds Test | Value | Signif. | I (0) | I (1) |
| F-statistic | 8.104871 *** | 10% | 1.85 | 2.85 |
| K | 8 | 5% | 2.11 | 3.15 |
| | | 2.5% | 2.33 | 3.42 |
| | | 1% | 2.62 | 3.77 |

Note: *** denote statistically significant at the 1% levels, respectively. The decisions are taken based on the critical values of Banerjee et al. (2017).

The present study investigates the long-run effect of FRI, RENN, TRO, and EGR on $CO_2E$ in Norway. Upon meeting the prerequisite conditions, an NARDL is used in the current study. In Table 5, we display the NARDL's long-run results. According to the results, financial stability causes a reduction in $CO_2$ emissions in Norway. A 1% expansion in financial stability lessens the pollution of the environment by 0.306% and is statistically significant. This long-run proof suggests that financial stability is good for the environment. In other words, positive and negative shocks to financial stability will have a beneficial impact on Norway's $CO_2$ emissions. The outcomes are in line with our expectations and theory, especially because having a stable financial system involves efficiently allocating resources, assessing and managing financial risks, ensuring that employment levels are close to the level of the natural rate of unemployment, and eliminating changes in price levels of real or financial assets which affect monetary stability or employment levels. The ability of a financial system to dissipate financial imbalances that arise as a result of adverse and unforeseen events is what we refer to as stability. The stability of the system will prevent adverse events from disrupting the real economy and other financial systems, as the system will absorb shocks through self-correcting mechanisms. A strong financial system is essential for economic growth since the bulk of transactions in the real economy is carried out on financial markets. When financial stability is absent, the real value of the institution can be best appreciated. During these times, banks are unlikely to finance profitable projects, asset prices diverge significantly from their intrinsic value, and payments may be delayed. Major instability can lead to bank runs, hyperinflation, or a stock market crash. It can severely shake confidence in the financial and economic system. Furthermore, during this time, when the financial system is unstable, banks are not confident to release funds for innovative technology that can combat carbon emissions. Secondly, [59] indicates that banks may hesitate to fund green technology if it involves assets that have intangible attributes and are linked to human capital. It is difficult to redeploy such assets and consequently hard to collateralize them [60]. Intangible assets and uncertainty are significant concerns for startups in the energy technology sector [61].

**Table 5.** Nonlinear-ARDL Long Run Form.

| Variable | Coefficient | Std. Error | t-Statistic | Prob. |
|---|---|---|---|---|
| FRI_POS | −0.306617 *** | 0.081949 | −3.741570 | 0.0004 |
| FRI_NEG | −0.063510 * | 0.036574 | −1.736482 | 0.0869 |
| EGR_POS | 2.829711 *** | 0.902205 | 3.136439 | 0.0025 |
| EGR_NEG | 16.58350 ** | 7.415659 | 2.236281 | 0.0286 |
| RENN_POS | −1.178169 *** | 0.361015 | −3.263494 | 0.0017 |
| RENN_NEG | −0.336383 ** | 0.142563 | −2.359542 | 0.0211 |
| TRO_POS | −0.879127 * | 0.453962 | −1.936565 | 0.0569 |
| TRO_NEG | −5.043676 ** | 2.203409 | −2.289033 | 0.0251 |
| C | 1.568700 *** | 0.016248 | 96.54816 | 0.0000 |
| CointEq (−1) * | −0.200395 *** | 0.020936 | −9.571848 | 0.0000 |
| | | Diagnostic Test | | |
| | Heteroskedasticity Test: Breusch-Pagan-Godfrey | | | |
| F-statistic | 1.289038 | Prob. F(25,69) | | 0.2031 |
| | Breusch-Godfrey Serial Correlation LM Test: | | | |
| F-statistic | 0.980167 | Prob. F(2,67) | | 0.3806 |

Note: *, ** and *** denote statistically significant at the 10%, 5%, and 1% levels, respectively.

Further, economic growth causes carbon emissions to climb according to empirical findings using an NARDL, in which a 1% increase in GDP magnifies the pollution in the environment by 2.8297%, and is statistically significant. In Norway, higher growth is likely to result in higher $CO_2E$. This means that a positive growth shock will have a statistically significant detrimental effect on Norwegian emissions, causing them to increase by 2.829%. A booming economy would be detrimental to Norway's environmental performance. However, the country has generally maintained good economic and social outcomes. Over time, the country was one of the top performers among OECD countries in GDP per capita. While post-pandemic economic adjustments, aging populations, and climate change are all posing challenges, good outcomes cannot be sustained.

Renewable energy has been favorable for $CO_2E$ in Norway. Positive and negative shocks in renewable energy have been beneficial to the environment in the case of Norway. A positive shock in renewable energy has reduced emissions by −1.17%, and this is statistically significant. Additionally, a negative shock to renewable energy in Norway over the long run would cause an increase in emissions by 0.33%. The outcome of our empirical study is consistent with novel research studies, such as the study that stated that sustained energy consumption lowers the $CCO_2$ emissions in Chile; it is also consistent with the findings of [62] that sustainable renewable energy mitigates $CO_2E$ and improves the quality of the environment. Renewable energy is becoming an increasingly important topic of discussion in some debates regarding the rebalancing of environmental and economic condition. With regard to climate and environmental issues, Norway has been a pioneer in many respects, including its exports of oil and gas, which contribute to its large emissions. At the same time, Norway has reduced emissions in many ways, among which are the widespread use of electric vehicles and its extensive hydropower capabilities. The country's energy supply is made up of 50 percent renewable energy. In order to further reduce emissions, Norway should engage in research and development, as well as adopt innovative technologies.

The results are presented in Table 6. The results showed that all indicators, FRI, TRO, RENN, and EGR, can predict $CO_2E$ in the NORWAY. Initiatives focused on FRI, TRO, RENN, and EGR will lower $CO_2E$ significantly. Furthermore, the J-B analysis indicates that the data are normal. Figures 1 and 2 show CUSUM and CUSUMSQ as well as stable models. Next, we assess the causal effect of FRI, TRO, RENN, and EGR on $CO_2E$ in Norway using the frequency domain causality approach developed by Breitung and Candelon (2006).

**Table 6.** Frequency domain causality test of Breitung and Candelon (2006).

| Direction of causality | Long Term | | Medium Term | | Short Term | |
|---|---|---|---|---|---|---|
| | $\omega_i = 0.01$ | $\omega_i = 0.05$ | $\omega_i = 1.00$ | $\omega_i = 1.50$ | $\omega_i = 2.00$ | $\omega_i = 2.50$ |
| FRI $\rightarrow$ CO$_2$E | < 8.719 > ** (0.012) | < 8.650 > ** (0.013) | < 0.226 > (0.893) | < 4.954 > * (0.084) | < 0.330 > (0.847) | < 1.735 > (0.419) |
| RENN $\rightarrow$ CO$_2$E | <1.391> (0.498) | <1.352> (0.508) | <3.089> ** (0.213) | <8.871> ** (0.011) | <7.909> ** (0.019) | <10.699> *** (0.004) |
| EGR $\rightarrow$ CO$_2$E | <6.936> ** (0.031) | <6.785> ** (0.033) | <2.792> (0.247) | <4.837> * (0.089) | <3.851> (0.145) | <2.050> (0.358) |
| TRO $\rightarrow$ CO$_2$E | <7.241> ** (0.026) | <7.488> ** (0.023) | <6.699> ** (0.035) | <6.333> ** (0.042) | <4.234> (0.120) | <6.063> ** (0.048) |

Note: <> and () stands for Wald test statistic and *p*-value, respectively. The path of causality is represented by $\rightarrow$. 10%, 5%, and 1% levels of significance are illustrated by *, **, and ***, correspondingly. SIC is used to verify the VAR model's lag lengths.

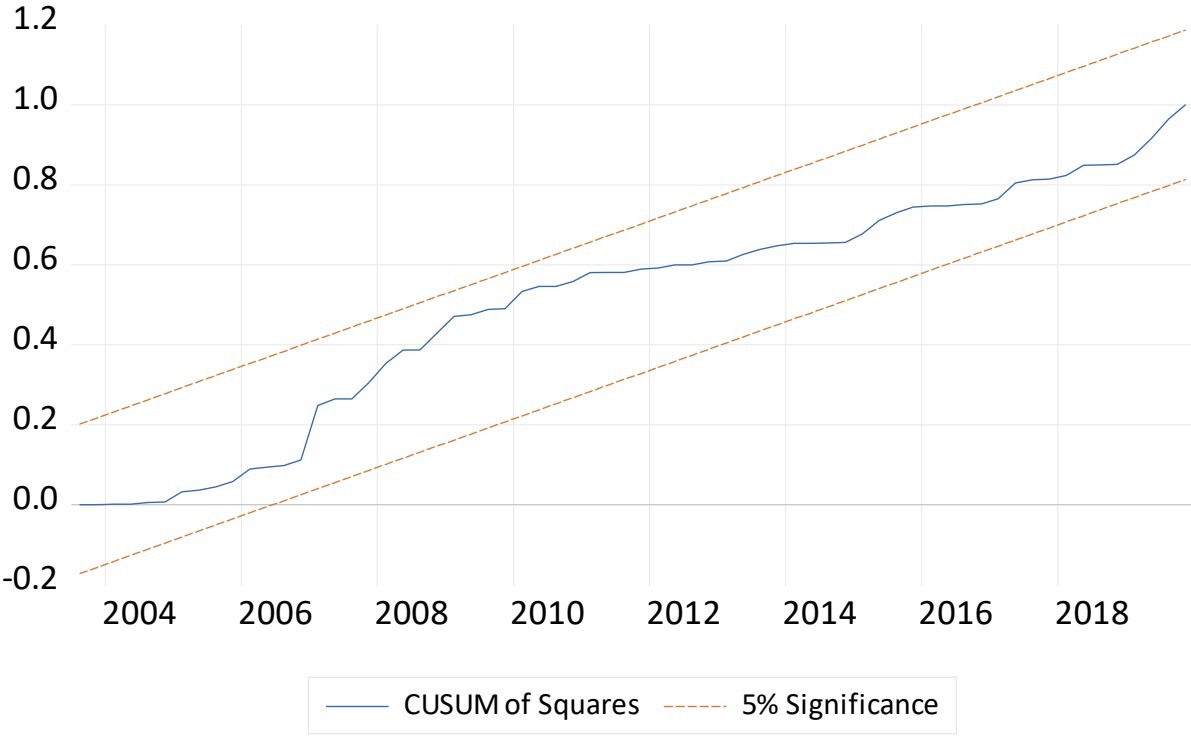

**Figure 1.** CUSUM of Squares.

Table 6 reports the outcomes of the frequency domain causality test of Breitung and Candelon (2006) for the estimated model in the present study. In other words, the table reports the causal effects of FRI, TRO, EGR, and RENN on CO$_2$E in Norway in the long term, medium term, and short term. Table 6 shows that the null hypothesis that FRI does not cause CO$_2$E can be rejected in the long term and medium term in Norway, implying that changes in financial stability in Norway lead to a change in environmental degradation. Moreover, EGR causes CO$_2$E in the long term and medium term, indicating that economic growth is a predictor of environmental degradation. Moreover, in the short term and medium term, RENN causes CO$_2$E in Norway. Finally, TRO Granger causes CO$_2$E at different frequencies. This implies that TRO is an important predictor of CO$_2$E in Norway in the short term, medium term, and long term. This illustrates that TRO can predict significant variation in CO$_2$E.

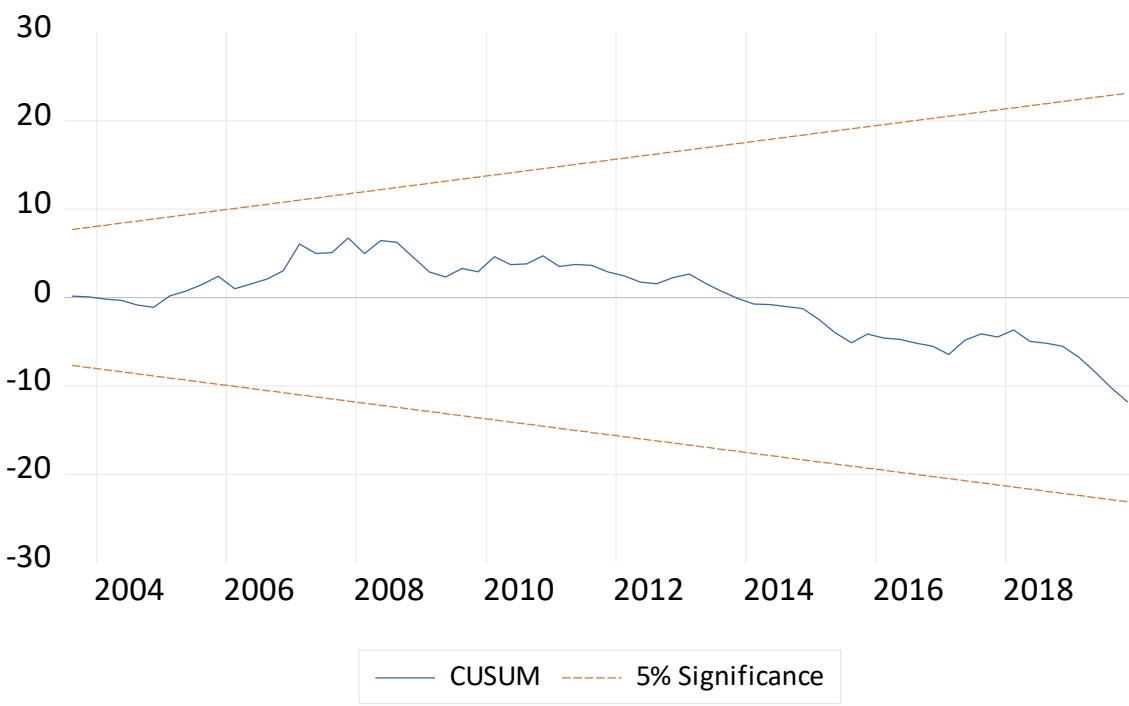

**Figure 2.** CUSUM.

### 4. Conclusions and Policy Direction

The objective of this study was to explore the effect of financial stability and carbon emissions while controlling for renewable energy, trade openness, and economic growth in Norway using annual data from 1995 to 2019. Background information on the strength of Norway's financial sector reveals that Norway's banking market is dominated by a few very large commercial banks, some regional-based banks, and several small savings banks. Further, Norwegian banks are resilient. They have access to funding and satisfy liquidity requirements by a good margin. Banks are solvent and fulfill capital requirements by a solid margin. In addition, banks' profitability has normalized over the past year, primarily owing to lower credit losses. As a result of the war in Ukraine, commodity prices have soared, particularly oil and gas. It brings in extra revenue. Norway has increased its carbon emissions in recent years because of its exports. Apart from a few studies that examined the effect of financial instability on environmental decline, few research studies have utilized financial risk as a proxy for financial stability. The empirical analysis is carried out by applying a nonlinear ARDL bound testing approach to cointegration. Cointegration relationships were confirmed by a bounds F-test of Norway's financial stability, economic growth, sustainability, trade openness, and $CO_2E$. Our result shows a negative and statistically significant relationship between financial stability and the deterioration of the ecosystem in Norway, providing evidence that healthy financial sectors are crucial for environmental quality improvements in Norway.

Based on the empirical evidence presented above, the current study makes the following policy recommendations.

- Norway is in a prime position to lead the way toward reducing its reliance on fossil fuels. Having relatively good economic standing, Norway may be in a better position than many other countries to switch to renewable energy from fossil fuel production because it is one of the wealthiest nations globally. Policymakers in Norway should first examine in detail the effects of global emissions on the economy under a wide range of scenarios for future climate policy so that they can make informed decisions about national petroleum policy in the long run. Further, there should be a robust energy policy in Norway. A climate test should be used by Norway in order to ensure

that planned oil production is consistent with climate goals in order to bridge the climate and energy policy gap.

- The financial sector should step up to support and lead the transition to a low-carbon economy. A lot of attention is being paid to evolving approaches, technologies, and methods that can be used to quantify how much lending and investment activities contribute to emissions. Using this study's findings, it is recommended that Norway concentrates on creating a stable financial system, which will encourage companies to adopt more advanced and efficient technologies, and in turn, will help to decrease energy consumption and improve the environment. Policymakers should encourage firms to adopt eco-friendly technologies and give them incentives to improve environmental quality through the financial sector. Obtaining a financial benefit will motivate firms to adopt environmentally friendly technologies, resulting in a reduction of energy consumption and carbon emissions.
- Moreover, regulations should be put in place that limit the availability of loans for businesses that discharge more waste into the water and air. Furthermore, in order to minimize carbon emissions caused by economic growth, Norway should pay attention to domestic consumers and energy-intensive industries. The switch to green energy is vital for rapid and cost-effective climate action. Since Norway has a robust economy, it should invest in renewable energy technologies with low emissions and impact on the environment, such as wind, solar, and hydro.
- To reduce emissions from the transport sector, key policies including carbon taxation, biofuel quotas, and requirements for using zero and low-emission technologies, as well as investment support schemes, must be strengthened.

**Author Contributions:** Conceptualization, D.K.; methodology, D.K. and M.O.O.; software, M.O.O. and R.A.C.; validation, S.Y.G.; formal analysis, S.Y.G.; investigation; M.O.O., resources, R.A.C.; data curation, M.O.O.; writing—original draft preparation, M.O.O.; writing—review and editing D.K.; G.C. and R.A.C.; visualization, D.K.; supervision, R.A.C.; project administration, S.Y.G. and G.C. All authors have read and agreed to the published version of the manuscript.

**Funding:** This paper is financed by Portuguese national funds through FCT–Fundação para a Ciência e a Tecnologia, I.P., project number UIDB/00685/2020. Also, The project is funded under the program of the Minister of Science and Higher Education titled "Regional Initiative of Excellence" in 2019–2022, project number 018/RID/2018/19, the amount of funding PLN 10 788 423,16.

**Institutional Review Board Statement:** Not applicable.

**Informed Consent Statement:** Not applicable.

**Data Availability Statement:** The variables used in this paper are collected from the database of World Bank and OECD.

**Conflicts of Interest:** The authors declare no conflict of interest.

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
