# Peer review of "The Asymmetric and Long-Run Effect of Financial Stability on Environmental Degradation in Norway"

_sustainability, doi:10.3390/su141610131_

Round 1
Reviewer 1 Report
1. Overall quality is good, however, few issues need to be fixed before publication process. My comments and suggestions are following:
2. Why specific period was selected and how?
3. Page 1, lines 26-28, do not mention only but be specific which policy?
4. Empirical analysis contains results? If yes, then it should be named as Results and discussions. Otherwise break them into two sections.
5. Many formatting issues in the paper. Tables are scattered and also of different styles. Keep the symmetry.
6. Figure 1 is immature. Better delete it.
7. Need to add more figures, as of now, this paper focuses too much on statistics.
8. Few typos exist in this paper, for instance, page 8, first paragraph.
9. Length of the paper can be reduced if possible. Especially in first two sections.
10. Do not provide a separate literature review section as it is not an economic study, merge it with the introduction, write story like and explain as a complete and thorough case.
11. Clear problem statement and rationale should be improved.
12. Look for redundancy of the references if any.
Author Response
Responses to the Comments
We are extremely thankful to respected Editors and reviewers for their interest in our work. The insightful comments of the reviewers improved our manuscript. They have brought up some good points, and we appreciate the opportunity to clarify our article. As indicated below, we have checked all the general, and specific comments provided by the reviewers and have made necessary changes accordingly to their indications.
The reviewer's suggestions and comments are in black, while our responses are in Red. All the necessary changes have been made in the updated manuscript via the track changes option.
- Overall quality is good, however, few issues need to be fixed before publication process. My comments and suggestions are following:
- Why specific period was selected and how?
Response: According to the above valuable comment from the reviewer, we have incorporated the above comment .in the methodology section,
- Page 1, lines 26-28, do not mention only but be specific which policy?
Response: We have incorporated the above comment
- Empirical analysis contains results? If yes, then it should be named as Results and discussions. Otherwise break them into two sections.
Response: we convert the title of section 4 to “Empirical Finding and Discussion”
- 5. Many formatting issues in the paper. Tables are scattered and also of different styles. Keep the symmetry.
Response: We fixed it but while uploading leading this issue, we will edit at the end of the paper will be accepted. Thanks a lot for this comment
- Figure 1 is immature. Better delete it.
Response: According to the above valuable comment from the reviewer, we have deleted the table.
- Need to add more figures, as of now, this paper focuses too much on statistics.
Response: Thank you very much for raising the valid point. We have two figure which are empirical proof for the estimator. The present study estimates the asymmetric and long-run effect of financial stability on environmental degradation in Norway, that is why it is a bit technical.
- 8. Few typos exist in this paper, for instance, page 8, first paragraph.
Response: Thank you very much for raising the valid point. However it has been fixed
- Length of the paper can be reduced if possible. Especially in first two sections.
Response: Thank you very much for valid point.The length has been reduced.
- Do not provide a separate literature review section as it is not an economic study, merge it with the introduction, write story like and explain as a complete and thorough case.
Response: Thank you very much for valid point. The length has been reduced.
- Clear problem statement and rationale should be improved.
Response: Thank you very much for valid point., problem statement has been improved.
- Look for redundancy of the references if any
Response: Thank you very much for the valid point, we re edit references in this paper

Reviewer 2 Report
Authors need to present theoretical underpinning.
Authors need to organize lit review with sub title.
Authors need to present research hypotheses.
Authors need to present analytic method and why it is adequate. Focusing on econometric method.
Theoretical contribution needs to be strengthened.
Author Response
Responses to the Comments
We are extremely thankful to respected Editors and reviewers for their interest in our work. The insightful comments of the reviewers improved our manuscript. They have brought up some good points, and we appreciate the opportunity to clarify our article. As indicated below, we have checked all the general, and specific comments provided by the reviewers and have made necessary changes accordingly to their indications.
The reviewer's suggestions and comments are in black, while our responses are in Red. All the necessary changes have been made in the updated manuscript via the track changes option.
1.Authors need to present theoretical underpinning.
Response: Thank you very much for raising the valid point.However it has been fixed
2.Authors need to organize lit review with sub title.
Response: Thank you very much for raising the valid point. However we have merged the introduction with Literature review
3.Authors need to present research hypotheses.
Response: According to the above valuable comment from the reviewer, we have incorporated the above comment .
4.Authors need to present analytic method and why it is adequate. Focusing on econometric method.
Response: According to the above valuable comment from the reviewer, we have incorporated the above comment. As requested we present analytic method and why it is adequate.

Reviewer 3 Report
sustainability-1823116
Manuscript ID: sustainability-1823116
Type of manuscript: Article
Title: The asymmetric and long-run effect of financial stability on
environmental degradation in Norway
The paper picked an important issue to address the asymmetric and long-run effect of financial stability on environmental degradation in Norway. The paper is still in the early stage, and not ready to be published. There are several suggestions to the authors for their revision.
1. The section of introduction is should be revised. The current status is incomplete and quite rough.
2. To increase the soundness of the view point in the paper, the reasoning should (1) include corresponding literature or (2) cite the sources of the specific statement. This is quite important. A research paper should be totally supported by research outcomes or reality facts. All of this should have improved in the sectors of introduction, literature review, and methodology.
3. The authors are suggested to rewrite the sector of introduction. The current section of introduction needs to be combined into the section of literature review. Please be sure that all statement should include their literature sources.
4. A sector of introduction should include (1) the background of the analysis, (2) the issue and the problem to be addressed, (3) the objectives that can be approached in this analysis, (4) analysis method applied, (5) the sector arrangement.
5. The title and the analysis is on Norway. However, in the current sections of introduction and literature review have barely mentioned the financial system, the financial stability, the environmental condition/degradation, the carbon dioxide emissions regarding the countries of Norway. The authors should focus on the targeted countries and the targeted issues in the writing of the sectors of introduction and literature review.
6. The interpretation of the research results and their discussion and conclusion should be based on the issues results, associated with the background, the problems/issues, and the literature review.

Author Response
Responses to the Comments
We are extremely thankful to respected Editors and reviewers for their interest in our work. The insightful comments of the reviewers improved our manuscript. They have brought up some good points, and we appreciate the opportunity to clarify our article. As indicated below, we have checked all the general, and specific comments provided by the reviewers and have made necessary changes accordingly to their indications.
The reviewer's suggestions and comments are in black, while our responses are in Red. All the necessary changes have been made in the updated manuscript via the track changes option.
Title: The asymmetric and long-run effect of financial stability on environmental degradation in Norway
The paper picked an important issue to address the asymmetric and long-run effect of financial stability on environmental degradation in Norway. The paper is still in the early stage, and not ready to be published. There are several suggestions to the authors for their revision.
- The section of introduction is should be revised. The current status is incomplete and quite rough.
Response: According to the above valuable comment from the reviewer, the introduction has been revised
- To increase the soundness of the view point in the paper, the reasoning should (1) include corresponding literature or (2) cite the sources of the specific statement. This is quite important. A research paper should be totally supported by research outcomes or reality facts. All of this should have improved in the sectors of introduction, literature review, and methodology.
Response: According to the above valuable comment from the reviewer, we have incorporated the above comment .
- The authors are suggested to rewrite the sector of introduction. The current section of introduction needs to be combined into the section of literature review. Please be sure that all statement should include their literature sources.
Response: According to the above valuable comment from the reviewer, we have incorporated the above comment .
- A sector of introduction should include (1) the background of the analysis, (2) the issue and the problem to be addressed, (3) the objectives that can be approached in this analysis, (4) analysis method applied, (5) the sector arrangement.
Response: According to the above valuable comment from the reviewer, this issue has been fixed
- The title and the analysis is on Norway. However, in the current sections of introduction and literature review have barely mentioned the financial system, the financial stability, the environmental condition/degradation, the carbon dioxide emissions regarding the countries of Norway. The authors should focus on the targeted countries and the targeted issues in the writing of the sectors of introduction and literature review.
Response: According to the above valuable comment from the reviewer, this issue has been fixed
- The interpretation of the research results and their discussion and conclusion should be based on the issues results, associated with the background, the problems/issues, and the literature review.
Response: Thanks a lot for this comments we fixed our paper based on this suggestion.

Round 2
Reviewer 3 Report
The paper is already well revised from the early draft. It looks ready to bee published.
This manuscript is a resubmission of an earlier submission. The following is a list of the peer review reports and author responses from that submission.